# Impact of COVID-19 Pandemic on Management and Outcomes in Patients with Septic Shock in the Emergency Department

**DOI:** 10.3390/jpm12111803

**Published:** 2022-11-01

**Authors:** Daun Jeong, Gun Tak Lee, Jong Eun Park, Tae Gun Shin, Kyunga Kim, Doeun Jang, Won Young Kim, You Hwan Jo, Sung Phil Chung, Jin Ho Beom, Sung-Hyuk Choi, Woon Yong Kwon, Gil Joon Suh, Byuk Sung Ko, Kap Su Han, Jong Hwan Shin, Hanjin Cho, Sung Yeon Hwang

**Affiliations:** 1Department of Emergency Medicine, Samsung Medical Center, Sungkyunkwan University School of Medicine, Seoul 06351, Korea; 2Department of Digital Health, Samsung Advanced Institute for Health Sciences & Technology, Sungkyunkwan University, Seoul 06355, Korea; 3Biomedical Statistics Center, Research Institute for Future Medicine, Samsung Medical Center, Seoul 06351, Korea; 4Department of Emergency Medicine, Asan Medical Center, University of Ulsan College of Medicine, Seoul 05505, Korea; 5Department of Emergency Medicine, Seoul National University Bundang Hospital, Seongnam 13620, Korea; 6Department of Emergency Medicine, Yonsei University College of Medicine, Seoul 03722, Korea; 7Department of Emergency Medicine, Guro Hospital, Korea University Medical Center, Seoul 08308, Korea; 8Department of Emergency Medicine, Seoul National University College of Medicine, Seoul 03080, Korea; 9Department of Emergency Medicine, Hanyang University College of Medicine, Seoul 04763, Korea; 10Department of Emergency Medicine, Korea University Anam Hospital, Seoul 02841, Korea; 11Department of Emergency Medicine, Seoul National University Boramae Medical Center, Seoul 07061, Korea; 12Department of Emergency Medicine, Korea University College of Medicine, Korea University Ansan Hospital, Ansan 15355, Korea

**Keywords:** COVID-19, septic shock, resuscitation, sepsis bundle, mortality

## Abstract

This study aimed to determine the impact of modifications in emergency department (ED) practices caused by the coronavirus disease 2019 (COVID-19) pandemic on the clinical outcomes and management of patients with septic shock. We performed a retrospective study. Patients with septic shock who presented to the ED between 1 January 2018 and 19 January 2020 were allocated to the pre-COVID-19 group, whereas those who presented between 20 January 2020 and 31 December 2020 were assigned to the post-COVID-19 group. We used propensity score matching to compare the sepsis-related interventions and clinical outcomes. The primary outcome measure was in-hospital mortality. Of the 3697 patients included, 2254 were classified as pre-COVID-19 and 1143 as post-COVID-19. A total of 1140 propensity score-matched pairings were created. Overall, the in-hospital mortality rate was 25.5%, with no statistical difference between the pre- and post-COVID-19 groups (*p* = 0.92). In a matched cohort, the post-COVID-19 group had delayed lactate measurement, blood culture test, and infection source control (all *p* < 0.05). There was no significant difference in time to antibiotics (*p* = 0.19) or vasopressor administration (*p* = 0.09) between the groups. Although sepsis-related interventions were delayed during the COVID-19 pandemic, there was no significant difference in the in-hospital mortality between the pre- and post-COVID-19 groups.

## 1. Introduction

Coronavirus disease 2019 (COVID-19), caused by the severe acute respiratory syndrome coronavirus 2 (SARS-CoV-2), has led to the rapid global spread of the viral disease, prompting the World Health Organization to declare a pandemic on 11 March 2020. COVID-19 is responsible for a significant increase in morbidity and mortality. Owing to the exponential spread of the disease, the entire healthcare system, including the emergency department (ED), has been severely affected and overwhelmed. Numerous efforts have been made to prevent the spread of SARS-CoV-2 in EDs, including stringent infection control measures, revised triage, and modified diagnostic and therapeutic procedures [1,2].

COVID-19 provides another significant hazard, in addition to the primary burden such as the morbidity and mortality that is associated with the viral infection [3,4]. During the pandemic era, emergency medical attention and resources have focused on identifying and treating possible COVID-19 patients [5]. There have been concerns that these strategies in EDs may have worsened acute care for other critical illnesses such as sepsis and septic shock. Sepsis and septic shock, potentially life-threatening medical emergencies, require prompt, labor-intensive interventions in the initial phase of ED presentation [6,7,8]. However, combined with the modified ED care process, the shortage of space and skilled workforce, caused by the allocation of medical resources to manage patients with suspected or confirmed COVID-19, may have impeded timely sepsis-related interventions in the ED, such as antibiotic administration, and compromised patient safety [9,10].

The challenges in the ED that arose during the initial treatment of time-sensitive illnesses, such as stroke and acute myocardial infarction, as a result of the COVID-19 pandemic, have been emphasized previously [1,11]. However, there are limited studies on the impact of the pandemic on the early management and outcomes of sepsis and septic shock patients. In the present study, we aimed to evaluate the impact of the changes in ED care processes brought about by the COVID-19 pandemic on the clinical outcomes and treatment of patients with septic shock.

## 2. Materials and Methods

We performed a retrospective study using data from the multicenter registry of the Korean Shock Society (KoSS). The KoSS was founded in 2014 to improve the research and management of patients presenting to the ED with sepsis or septic shock [12]. The KoSS registry is an ongoing prospective multicenter study that includes 14 hospitals in South Korea. This registry is for consecutive patients who meet the following criteria within 6 h after ED presentation: (1) age >18 years, (2) persistent hypotension or hyperlactatemia, and (3) suspected or confirmed infection. Persistent hypotension (MAP < 70 mmHg; systolic BP < 90 mmHg) after adequate fluid treatment (20–30 mL/kg crystalloid solution) was characterized as refractory hypotension [13,14,15,16]. A blood lactate concentration ≥4 mmol/L was characterized as hyperlactatemia [17]. Patients who met the following criteria were excluded: (1) those who had previously signed a “Do not attempt resuscitation” order, (2) those who had documented persistent hypotension or hyperlactatemia 6 h after ED arrival, (3) those who were transferred from other hospitals without meeting the inclusion criteria upon ED arrival, and (4) those who were directly transferred from EDs to other hospitals. Data were collected in a standardized format in a web-based registry. The designated investigator reviewed and monitored the data on a periodic basis to ensure high quality. All patients were treated according to the updated international guidelines, which included initial fluid management, administration of broad-spectrum antibiotics, vasopressors, steroids, early source control, and general critical care. The institutional review boards of the participating centers approved this study, and written informed consent was obtained from the patients or their legal representatives prior to data collection.

### 2.1. ED Process after COVID-19

ED process modifications following the Korea Centers for Disease Control and Prevention (KCDC) guidelines have been implemented across institutions to prevent transmission of the virus in the ED. In brief, emergency medical personnel donned personal protective equipment (PPE), including a full bodysuit or at least a waterproof gown, gloving, shoe covers, goggles, N95 filtering facepiece respirators or its equivalents, and powered air-purifying respirators, as appropriate [18]. Before ED entrance, triage or tele-triage was performed at independent facilities to separate high-suspicion patients from low-suspicion patients, while stratifying them according to illness severity. A patient with suspected COVID-19 was isolated in an isolation room that was negatively pressurized with its own ventilation and airflow system. With a few exceptions, advanced imaging or interventions for isolated patients were delayed until the COVID-19 test results were negative. There were differences in the available resources such as space and staffing at each local institution. Consequently, while participating institutions generally adhered to the KCDC’s guidelines for in-hospital infection prevention, each institution had to respond optimally to the specific scenario it encountered.

### 2.2. Study Population and Data Extraction

Data from the KoSS registry between January 2018 and December 2020 were abstracted for this study. According to the KCDC, the first COVID-19 patient in Korea was confirmed on 20 January 2020. We categorized patients into two groups based on the date of their ED presentation: pre-COVID-19 group (1 January 2018 to 19 January 2020) and post-COVID-19 group (20 January 2020 to 31 December 2020) [19]. The following data were retrieved from the registry: demographic characteristics, including age and sex; comorbidities, including diabetes, metastatic cancer, chronic liver disease, cardiac disease, chronic lung disease, chronic kidney disease, and chronic kidney disease on hemodialysis; vital signs upon ED arrival; laboratory data, including white blood cell count, hemoglobin, platelet count, PT, creatinine, albumin, and lactate; severity of disease; APACHE II (Acute Physiology and Chronic Health Evaluation II) score, Sequential Organ Failure Assessment (SOFA) score, ICU admission, mechanical ventilation support, and renal replacement therapy use; infection focus; sepsis-related interventions, including blood culture, antibiotics, initial fluid resuscitation, vasopressor use, and source control; and outcome-related data, including mortality and length of stay. The time frame for performing sepsis-related interventions was determined based on the patient’s arrival time at the ED as a point of reference. Compliance with fluid resuscitation was defined as intravenous administration of at least 30 mL/kg of crystalloid fluid within 3 h of ED arrival.

### 2.3. Outcome Measures

The primary outcome was in-hospital mortality rate. Secondary outcomes were 28-day mortality, ED length of stay (ED-LOS), and ED-LOS for ICU-admitted patients.

### 2.4. Statistical Analyses

Continuous variables are described as means with standard deviations or medians with interquartile ranges, depending on whether they satisfied the normality assumption (Shapiro–Wilk test). The pre- and post-COVID-19 groups were compared using either the independent t-test or the Wilcoxon rank sum test, as appropriate. Categorical variables were presented as frequencies with percentages, and the two groups were compared using the chi-square or Fisher’s exact test, as applicable. Propensity score (PS) matching was used to adjust for imbalances between the two groups. Age, sex, comorbidities, severity measures, initial lactate level, source of infection, and septic shock criteria meeting the Sepsis-3 consensus definition were used as the matching variables. We performed 1:1 matching using a caliper of 0.2. Balance was evaluated based on the mean standardized differences. Additionally, a multiple logistic regression analysis was conducted in the unmatched cohort to determine the variables associated with in-hospital mortality. The adjustment variables were chosen based on clinical plausibility, and the COVID-19 period was imposed on the model. The final models included the following variables: patient demographics, including age, sex, and comorbidities (metastatic cancer, chronic liver disease, and chronic lung disease); infection focus; initial lactate levels; initial SOFA score; Sepsis-3-defined septic shock; time to source control; time to antibiotic use; and time to vasopressor use. The adjusted odds ratios (aOR) with a 95% confidence interval (CI) were determined. Statistical significance was defined as a two-tailed *p*-value < 0.05. Statistical analysis was performed using SAS (version 9.4; SAS Institute, Cary, NC, USA) and R version 3.6.2 (Vienna, Austria; http://www.R-project.org / accessed on 1 December 2021) by independent biostatisticians.

## 3. Results

### 3.1. Baseline Characteristics

We analyzed the data of 3697 patients from a total of 3725 patients who were registered in the KoSS registry, excluding 28 patients whose initial lactate values were missing (Figure 1). There were 2254 and 1143 patients classified into the pre-COVID-19 and post-COVID-19 groups, respectively. From these patients, 1140 PS-matched pairs were generated (one-to-one matching: pre-COVID-19 group, *n* = 1140 vs. post-COVID-19 group, *n* = 1140). None of the patients tested positive for COVID-19 in the post-COVID-19 group. The baseline characteristics of the unmatched and propensity-matched cohorts are shown in Table 1. For matching variables, the PS matching approach resulted in balanced grouping.

### 3.2. Primary and Secondary Outcomes

The primary and secondary outcomes are shown in Table 2 and Appendix A. In the PS-matched cohort, the overall in-hospital mortality rate was 25.5% (*n* = 582/2280), and there was no significant difference between the pre- and post-COVID-19 groups (25.4%, *n* = 290/1140 vs. 25.6%, *n* = 292/1140, respectively, *p* = 0.92). Both the ED-LOS (8.9 h [IQR 5.7–18.8] vs. 11.4 h [IQR, 7.2–19.7], *p* < 0.001) and the ED-LOS for ICU-admitted patients (6.9 h [IQR, 5.0–10.4] vs. 8.8 h [IQR, 5.8–13.7], *p* < 0.001) were longer in the post-COVID-19 group than in the pre-COVID-19 group.

### 3.3. Sepsis-Related Intervention

Sepsis-related interventions are shown in Table 3 and Appendix A. Time to lactate measurement (27.0 min vs. 37.0 min), time to blood culture test (82.0 min vs. 117.0 min), and time to source control (12.9 h vs. 16.1 h) were significantly longer in the post-COVID-19 group compared to the pre-COVID-19 group in a matched cohort (all *p* < 0.05). There was no significant difference between the groups in terms of time to IV antibiotics (pre-COVID-19 group, 140.5 min vs. post-COVID-19 group, 138.0 min, *p* = 0.19) or vasopressor (pre-COVID-19 group, 133.0 min vs. post-COVID-19 group, 143.5 min, *p* = 0.09) administration. The pre-COVID-19 group had higher compliance with fluid resuscitation within 3 h than the post-COVID-19 group (74.9% vs. 68.3%, *p* < 0.001; Table 3, Figure 2).

### 3.4. Variables Associated with in-Hospital Mortality

The COVID-19 period was not found to be associated with in-hospital mortality in the univariable logistic regression analysis (OR, 1.14 [95% CI, 0.97–1.34], *p* = 0.11; Table 4). In the multivariable logistic regression analyses, the COVID-19 period was not associated with in-hospital mortality (aOR, 1.00 [95% CI, 0.84–1.20], *p* = 0.98). Initial lactate levels (aOR, 1.13 [95% CI, 1.10–1.16], *p* < 0.001), SOFA score (aOR, 1.18 [95% CI, 1.16–1.22], *p* < 0.001), and Sepsis-3-defined septic shock (aOR, 1.63 [95% CI, 1.30–2.04], *p* < 0.001) were associated with in-hospital mortality. Delayed antibiotic administration (≥3 h) was significantly associated with an increase in in-hospital mortality, compared to antibiotic administration within three hours.

## 4. Discussion

This study examined the impact of the COVID-19 pandemic on the clinical outcomes and management of patients with septic shock in EDs. In-hospital or 28-day mortality rates were not significantly different between the pre- and post-COVID-19 groups. Several sepsis-related interventions, including lactate measurement, blood culture testing, infection source control, and fluid resuscitation, were delayed in the post-COVID-19 group compared with the pre-COVID-19 group. Additionally, the ED-LOS and ED-LOS in ICU-admitted patients were prolonged in the post-COVID-19 group. This study is clinically relevant, as it indicates the impact of the COVID-19 pandemic on acute care in patients with septic shock in the ED.

Our study draws attention to potential challenges in the management of patients with septic shock during the COVID-19 pandemic, using previous years as a reference. Several sepsis-related interventions were delayed, and bundle compliance was reduced; prolonged ED-LOS and ED-LOS for ICU-admitted patients occurred during the COVID-19 pandemic compared with the pre-pandemic era. Our results are consistent with the findings of studies that reported significant in-hospital management delays for several time-sensitive illnesses during the COVID-19 pandemic. Katsanos et al. found an increase in the median door-to-CT time in patients receiving intravenous tissue plasminogen activator [20]. In addition, Bruoha et al. found substantial delays in door-to-ECG, ECG-to-balloon, and door-to-balloon times in patients with acute coronary syndrome [21]. Many infection prevention and control measures have been adopted in the ED to minimize viral transmission within hospitals, assuming that every septic patient entering the ED is potentially infected with COVID-19 [9,22]. Intensive screening during the ED triage, physical isolation, additional laboratory screening for COVID-19, postponement of ancillary tests to protect diagnostic services, and the physiological and psychological stress associated with wearing PPE might also result in communication disruption and decreased manual dexterity. All of the above factors may hinder the provision of adequate therapy and prolong the ED-LOS [21,22,23,24].

Our findings suggest that modifications to the ED care process during the COVID-19 pandemic do not significantly affect short-term mortality in critically ill septic patients. Few studies have assessed the impact of the COVID-19 pandemic on the outcomes of patients with septic shock who presented to the ED. In a retrospective single-center study (*n* = 216), Kim et al. reported no significant difference in the in-hospital mortality rate between the pre- and post-COVID-19 pandemic periods in patients with septic shock who presented to the ED [25]. Disease severity, indicated by the SAPS III score, was the only significant risk factor for in-hospital mortality (OR, 1.07; 95% CI 1.04–1.10; *p* < 0.001). However, because this was a single-center study with a small sample size, there are some limitations in generalizing the results. Our study was conducted in a multicenter setting with a large sample size. It included a broader spectrum of sepsis patients and ensured data quality by prospectively collecting predefined variables. Our results are consistent with those of a previous study. There were no significant differences in in-hospital mortality or 28-day mortality between the pre- and post-COVID-19 groups in the PS-matched cohort. Multivariable analysis of the unmatched cohort also showed that the COVID-19 period was not associated with in-hospital mortality. In contrast, well-known factors such as underlying disease, lactate level, and disease severity were significantly associated with in-hospital mortality. It is unclear why there was no difference in in-hospital mortality between the two groups; however, this might be explained as follows. First, despite the delay in blood culture testing during the pandemic, there was no difference in the timing of antibiotic administration between the two groups. There was also no difference in the timing of the administration of vasopressors. Both are critical components in the treatment of septic shock [6]. Even though lactate measurements were delayed, it did not appear to be clinically significant enough to impact the prognosis of the patient. Despite the fact that source control was delayed in the post-COVID-19 group, the multivariable analysis revealed that source control performed 12 h after arrival at the ED was not associated with an increased in-hospital mortality rate, compared to those who received source control within 12 h. Second, despite the fact that the severity index of patients during the COVID-19 pandemic period was worse than that of the control period, it did not appear to be severe enough to alter the outcomes. Finally, it appears that the number of ED patients declined during the pandemic, allowing medical professionals to devote more time and resources to treating critically ill patients.

This study did not provide any data on patients’ healthcare-seeking behaviors. However, a number of studies have shown that patients avoid or delay medical contact, possibly due to fear of contracting COVID-19 [26,27,28,29,30,31]. This may lead to increased severity of illness at the time of ED presentation. According to Venkatesh et al., the number of visits to ED declined dramatically in the initial phase of the pandemic. Although it recovered gradually thereafter, it remained 23% lower than in the pre-pandemic period [32]. In this study, the worsened severity index represented by a substantial increase in initial lactate level, initial SOFA score, APACHE II score, and frequency of septic shock, according to the Sepsis-3 definition, may imply delayed presentation to the ED among septic patients during the COVID-19 pandemic.

This study had several limitations. First, although the data were obtained from a prospective multicenter registry of consecutive patients, using a standardized and predetermined protocol, they were analyzed retrospectively. We performed a PS-matched analysis to minimize the inherent biases associated with the study design. However, nonidentifiable confounding factors may have affected the results, and it was difficult to determine the causal relationship between the variables and outcomes. Second, we were unable to assess the appropriateness of the initial care, such as inappropriate use of antibiotics. Finally, local institution conditions may have necessitated some adaptations to the KCDC’s guidelines for in-hospital infection control and prevention, despite the fact that participating institutions mainly adhered to these guidelines. However, these factors were not considered in this study.

## 5. Conclusions

In this study, the in-hospital mortality was not significantly different between the pre- and post-COVID-19 groups. Several sepsis-related interventions were delayed in the post-COVID-19 group. Our findings suggest that modifications to the ED care process during the COVID-19 pandemic did not significantly affect the clinical outcomes of patients with septic shock.

## Figures and Tables

**Figure 1 jpm-12-01803-f001:**
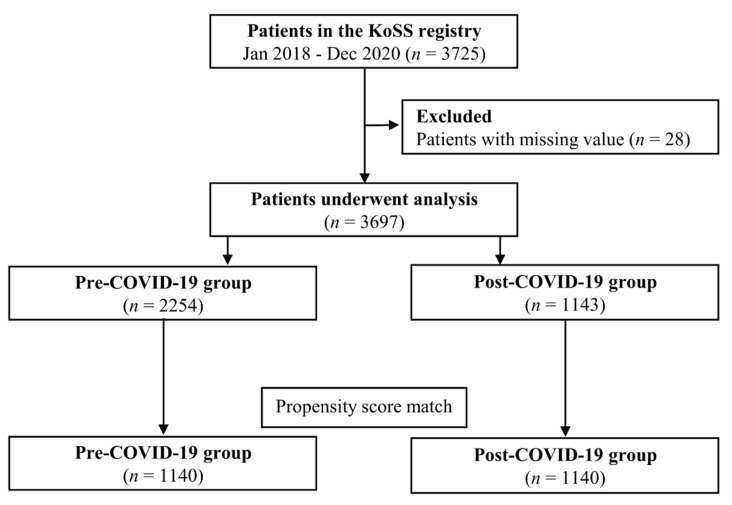
Study flowchart. KoSS, Korean Shock Society; COVID-19, coronavirus disease 2019.

**Figure 2 jpm-12-01803-f002:**
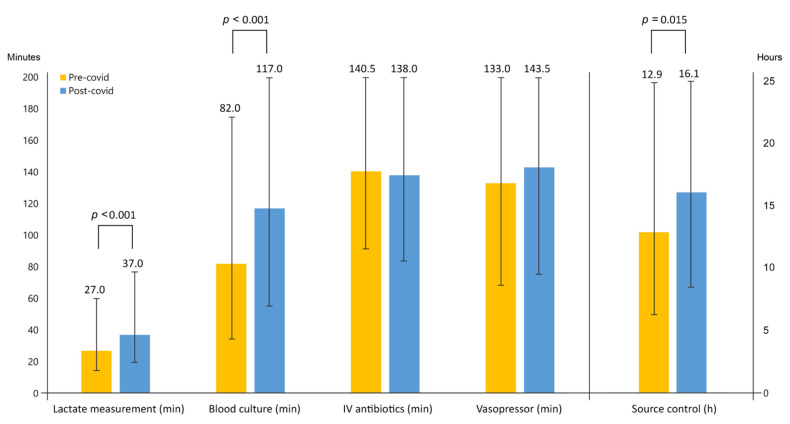
Propensity-matched cohort analysis of time interval from triage to sepsis shock-related intervention. The median time to lactate measurement in the pre-COVID-19 group and post-COVID-19 group was 27.0 min [IQR 14.0–60.0] and 37.0 min [IQR 20.0–77.0] (*p* < 0.05). The median time to blood culture in the pre-COVID-19 group and post-COVID-19 groups was 82.0 min [IQR 34.0–175.0] and 117.0 min [IQR 55.0–205.0] (*p* < 0.05). The median time to IV antibiotics in the pre-COVID-19 group and post-COVID-19 groups was 140.5 min [IQR 91.0–222.0] and 138.0 min [IQR 84.0–213.0] (*p* = 0.19). The median time to vasopressor in the pre-COVID-19 group and post-COVID-19 groups was 133.0 min [IQR 68.0–240.0] and 143.5 min [IQR 75.0–253.0] (*p* = 0.09). The median time to source control in the pre-COVID-19 group and post-COVID-19 groups was 12.9 h [IQR 6.3–27.4] and 16.1 h [IQR 8.6–49.1] (*p* = 0.01). COVID-19, coronavirus disease 2019; IV, intravenous.

**Table 1 jpm-12-01803-t001:** Baseline characteristics of unmatched and propensity-matched cohorts.

	Unmatched Cohort				Propensity-Matched Cohort			
	Overall (N = 3697)	Pre-COVID-19 (N = 2554)	Post-COVID-19 (N = 1143)	*p*-Value	Overall (N = 2280)	Pre- COVID-19 (N = 1140)	Post-COVID-19 (N = 1140)	*p*-Value
Age * (years)	70.4 [61.6–78.7]	70.6 [61.8–78.7]	69.85 [61.2–78.7]	0.57	70.3 [61.7–78.6]	70.7 [62.2–78.5]	69.82 [61.2–78.7]	0.85
Sex *, Male	2068 (55.9)	1436 (56.2)	632 (55.3)	0.60	1234 (54.1)	604 (53.0)	630 (55.3)	0.28
Comorbidity *								
Diabetes	1193 (32.3)	778 (30.5)	415 (36.3)	<0.001	819 (35.9)	407 (35.7)	412 (36.1)	0.80
Metastatic cancer	1124 (30.4)	749 (29.3)	375 (32.8)	0.033	745 (32.7)	372 (32.7)	373 (32.7)	0.96
Chronic liver disease	334 (9.0)	216 (8.5)	118 (10.3)	0.07	242 (10.6)	125 (11.0)	117 (10.3)	0.57
Cardiac disease	539 (14.6)	347 (13.6)	192 (16.8)	0.011	366 (16.1)	176 (15.4)	190 (16.7)	0.39
Chronic lung disease	306 (8.3)	197 (7.7)	109 (9.5)	0.06	214 (9.4)	107 (9.4)	107 (9.4)	1.00
Chronic kidney disease	343 (9.3)	225 (8.8)	118 (10.3)	0.14	228 (10.0)	111 (9.7)	117 (10.3)	0.68
Chronic kidney disease on hemodialysis	78 (2.1)	50 (2.0)	28 (2.4)	0.34	59 (2.6)	31 (2.7)	28 (2.5)	0.69
Laboratory data								
WBC (×10^3^/μL)	10.2 [4.6–16.7]	10.3 [5.0–16.7]	9.9 [3.8–16.5]	0.09	10.1 [4.4–16.8]	10.2 [5.1–17.1]	10.0 [3.8–16.6]	0.08
Hemoglobin (g/dl)	10.7 [9.1–12.5]	10.7 [9.2–12.4]	10.6 [8.9–12.6]	0.23	10.6 [8.9–12.4]	10.6 [9.0–12.3]	10.6 [8.9–12.6]	0.59
Platelet (×10^3^/μL)	143 [79–223]	147 [81–225]	135 [76.5–217]	0.029	141 [75–219]	143 [74–220.5]	135 [77–217]	0.34
PT, INR	1.25 [1.12–1.47]	1.24 [1.11–1.45]	1.26 [1.13–1.49]	0.10	1.26 [1.13–1.51]	1.26 [1.13–1.52]	1.26 [1.13–1.49]	0.22
Creatinine (mg/dl)	1.36 [0.92–2.19]	1.35 [0.93–2.14]	1.37 [0.91–2.30]	0.63	1.38 [0.93–2.31]	1.38 [0.95–2.32]	1.37 [0.91–2.31]	0.40
Albumin (g/dl)	3.0 [2.5–3.5]	3.0 [2.5–3.5]	3.0 [2.6–3.5]	0.025	3.0 [2.5–3.5]	3.0 [2.5–3.4]	3.0 [2.6–3.5]	0.002
Lactate * (mmol/L)	3.6 [2.0–5.6]	3.5 [2.0–5.5]	3.8 [2.2–5.8]	0.005	3.9 [2.2–5.9]	3.9 [2.2–5.9]	3.8 [2.2–5.8]	0.58
Procalcitonin (ng/mL)	7.60 [1.28–33.45]	7.16 [1.16–31.95]	8.76 [1.45–38.28]	0.024	8.19 [1.32–35.82]	7.28 [1.09–34.60]	8.76 [1.45–38.28]	0.10
CRP (mg/dl)	14.06 [5.69–23.72]	13.40 [5.08–23.31]	15.03 [6.91–24.56]	<0.001	13.90 [6.03–23.44]	12.72 [5.19–22.13]	15.04 [6.92–24.63]	<0.001
Resistant bacteria	329 (8.9)	243 (9.5)	86 (7.5)	0.049	213 (9.3%)	127 (11.1)	86 (7.5)	0.003
APACHE II score *	19 [14–26]	19 [14–26]	20 [15–26]	0.010	20 [14–26]	19.5 [14–26]	20 [15–26]	0.78
SOFA score *	8 [6–11]	8 [5–11]	8 [6–11]	0.003	8 [6–11]	8 [5–11]	8 [6–11]	0.88
Infection focus *								
Lung infection	1205 (32.6)	835 (32.7)	370 (32.4)	0.85	730 (32.0)	361 (31.7)	369 (32.4)	0.72
Urinary tract infection	1047 (28.3)	708 (27.7)	339 (29.7)	0.23	670 (29.4)	333 (29.2)	337 (29.6)	0.86
Gastrointestinal infection	645 (17.5)	424 (16.6)	221 (19.3)	0.043	444 (19.5)	226 (19.8)	218 (19.1)	0.67
Hepatobiliary infection	782 (21.2)	513 (20.1)	269 (23.5)	0.018	516 (22.6)	248 (21.8)	268 (23.5)	0.31
Other focus	481 (13.0)	354 (13.9)	127 (11.1)	0.022	267 (11.7)	140 (12.3)	127 (11.1)	0.38
Septic shock *†	2090 (56.5)	1414 (55.4)	676 (59.1)	0.032	1359 (59.6)	685 (60.1)	674 (59.1)	0.63
ICU admission	2126 (57.5)	1506 (59.0)	620 (54.2)	0.008	1306 (57.3)	687 (60.3)	619 (54.3)	<0.001
Ventilator support	1103(29.8)	753 (29.5)	350 (30.6)	0.48	707 (31.0)	358 (31.4)	349 (30.6)	0.68
RRT				0.040				0.020
None	3169 (85.7)	2195 (85.9)	974 (85.2)		1916 (84.0)	944 (82.8)	972 (85.3)	
Within 24 h	360 (9.7)	257 (10.1)	103 (9.0)		245 (10.8)	142 (12.5)	103 (9.0)	
After 24 h	168 (4.5)	102 (4.0)	66 (5.8)		119 (5.2)	54 (4.7)	65 (5.7)	
Source control	917 (24.8)	657 (25.7)	260 (22.8)	0.053	559 (24.5)	301 (26.4)	258 (22.6)	0.036

The data are presented as median (IQRs) for continuous variables or as numbers (%) for categorical variables. * Propensity score–matched variables. ^†^ Septic shock that meets the criteria defined by the Sepsis-3 consensus definition. COVID-19, coronavirus disease 2019; WBC, white blood cell; PT, prothrombin time; INR, international normalized ratio; CRP, C-reactive protein; APACHE II, Acute Physiology and Chronic Health Evaluation II; SOFA, Sequential Organ Failure Assessment; ICU, intensive care unit; RRT, renal replacement therapy.

**Table 2 jpm-12-01803-t002:** Primary and secondary outcomes.

	Unmatched Cohort				Propensity-Matched Cohort			
	Overall (N = 3697)	Pre-COVID-19 (N = 2254)	Post-COVID-19 (N = 1143)	*p*-Value	Overall (N = 2280)	Pre-COVID-19 (N = 1140)	Post-COVID-19 (N = 1140)	*p*-Value
In-hospital mortality	888 (24.0)	594 (23.3)	294 (25.7)	0.11	582 (25.5)	290 (25.4)	292 (25.6)	0.92
28-day mortality	849 (23.0)	578 (22.6)	271 (23.7)	0.42	550 (24.1)	280 (24.6)	270 (23.7)	0.30
ED LOS (h)	9.8 [6.1–19.1]	9.0 [5.8–18.8]	11.4 [7.2–19.7]	<0.001	10.2 [6.3–19.2]	8.9 [5.7–18.8]	11.4 [7.2–19.7]	<0.001
ED LOS for ICU admitted patients (h)	7.4 [5.2–11.4]	7.0 [5.0–10.3]	8.8 [5.8–13.7]	<0.001	7.7 [5.2–12.0]	6.9 [5.0–10.4]	8.8 [5.8–13.7]	<0.001

The data are presented as median (IQRs) for continuous variables or as numbers (%) for categorical variables. COVID-19, coronavirus disease 2019; ED, emergency department; LOS, length of stay; ICU, intensive care unit.

**Table 3 jpm-12-01803-t003:** Sepsis-related interventions of unmatched and propensity-matched cohorts.

Interventions *	Unmatched Cohort				Propensity Score-Matched Cohort			
	Overall (N = 3697)	Pre-COVID-19 (N = 2554)	Post-COVID-19 (N = 1143)	*p*-Value	Overall (N = 2280)	Pre-COVID-19 (N = 1140)	Post-COVID-19 (N = 1140)	*p*-Value
Time to lactate measurement (min)	30.0 [16.0–66.0]	27.0 [15.0–59.0]	37.0 [20.0–77.0]	<0.001	32.0 [17.0–68.0]	27.0 [14.0–60.0]	37.0 [20.0–77.0]	<0.001
Time to blood culture (min)	90.0 [38.0–187.0]	80.0 [33.0–175.0]	117.5 [55.0–206.0]	<0.001	97.5 [43.5–191.5]	82.0 [34.0–175.0]	117.0 [55.0–205.0]	<0.001
Blood culture within 1 h	1365 (37.4)	1050 (41.5)	315 (28.2)	<0.001	781 (34.8)	467 (41.4)	314 (28.2)	<0.001
Blood culture within 3 h	2692 (73.8)	1926 (76.2)	766 (68.5)	<0.001	1625 (72.4)	860 (76.2)	765 (68.6)	<0.001
Time to IV antibiotics (min)	140.0 [87.0–220.0]	140.0 [88.0–223.0]	139.0 [84.0–214.0]	0.17	139.0 [87.0–217.0]	140.5 [91.0–222.0]	138.0 [84.0–213.0]	0.19
IV antibiotics within 1 h	473 (12.9)	329 (13.0)	144 (12.6)	0.77	292 (12.9)	148 (13.1)	144 (12.6)	0.71
IV antibiotics within 3 h	2358 (64.1)	1612 (63.5)	746 (65.3)	0.28	1465 (64.5)	719 (63.4)	746 (65.5)	0.29
Fluid resuscitation within 3 h ^†^	2672 (72.3)	1894 (74.2)	778 (68.1)	<0.001	1632 (71.6)	854 (74.9)	778 (68.3)	<0.001
Time to vasopressor (min) ^‡^	143.0 [74.0–252.0]	143.0 [74.0–252.0]	143.5 [74.5–253.5]	0.66	138.0 [72.0–245.0]	133.0 [68.0–240.0]	143.5 [75.0–253.0]	0.09
Vasopressor within 1 h ^‡^	672 (20.4)	463 (20.3)	209 (20.7)	0.76	439 (21.5)	231 (22.3)	208 (20.7)	0.31
Vasopressor within 3 h ^‡^	1987 (60.3)	1391 (60.9)	596 (59.1)	0.34	1252 (61.3)	657 (63.5)	595 (59.2)	0.034
Time to source control (h) ^§^	14.5 [7.2–40.0]	13.4 [6.7–32.3]	16.4 [8.6–49.2]	0.002	14.7 [7.3–42.0]	12.9 [6.3–27.4]	16.1 [8.6–49.1]	0.015

The data are presented as median (IQRs) for continuous variables or as numbers (%) for categorical variables. * The time frame for performing sepsis-related interventions was determined using the patient’s arrival time at the emergency department (ED) as a point of reference. ^†^ Those who received intravenous administration of at least 30 mL/kg of crystalloid fluid within 3 h after ED arrival. ^‡^ Patients requiring vasopressors. ^§^ Patients requiring infection source control. COVID-19, coronavirus disease 2019; IV, intravenous.

**Table 4 jpm-12-01803-t004:** Univariable and multivariable logistic regression analysis of in-hospital mortality.

	Univariable		Multivariable	
	OR (95% CI)	*p*-Value	aOR (95% CI)	*p*-Value
COVID-19 period	1.14 (0.97–1.34)	0.11	1.00 (0.84–1.20)	0.98
Age, years	1.01 (1.00–1.02)	0.001	1.01 (1.01–1.02)	<0.001
Sex, male	0.85 (0.73–0.99)	0.035	1.12 (0.93–1.33)	0.23
Metastatic cancer	1.38 (1.17–1.62)	<0.001	1.54 (1.28–1.86)	<0.001
Chronic liver disease	1.44 (1.13–1.84)	0.004	1.10 (0.83–1.47)	0.50
Chronic lung disease	1.53 (1.18–1.97)	0.001	1.38 (1.03–1.86)	0.031
Infection focus				
Lung	Reference		Reference	
Urinary tract	0.34 (0.27–0.55)	<0.001	0.40 (0.31–0.52)	<0.001
Gastrointestinal tract	0.85 (0.68–1.07)	0.17	0.96 (0.73–1.25)	0.76
Hepatobiliary tract	0.36 (0.28–0.46)	<0.001	0.47 (0.35–0.64)	<0.001
Other focus	0.67 (0.67–0.53)	0.001	0.76 (0.57–0.99)	0.045
Lactate (+1 mmol/L)	1.20 (1.18–1.23)	<0.001	1.13 (1.10–1.16)	<0.001
SOFA score, initial	1.24 (1.21–1.26)	<0.001	1.19 (1.16–1.22)	<0.001
Septic shock *	2.94 (2.48–3.48)	<0.001	1.63 (1.30–2.04)	<0.001
Source control ^†^				
<12 h	Reference		Reference	
≥12 h	0.96 (0.67–1.38)	0.83	0.99 (0.67–1.46)	0.971
None	1.94 (1.47–2.56)	<0.001	1.77 (1.28–2.46)	0.001
Antibiotics ^†^				
<3 h	Reference		Reference	
≥3 h	0.96 (0.82–1.13)	0.65	1.21 (1.01–1.45)	0.038
Vasopressor ^†^				
<3 h	Reference		Reference	
≥3 h	0.77 (0.66–0.91)	0.002	1.02 (0.85–1.24)	0.812
None	0.57 (1.01–1.79)	<0.001	1.47 (1.02–2.12)	0.038

* Septic shock that meets the criteria defined by the Sepsis-3 consensus definition. ^†^ The time frame for performing sepsis-related interventions was determined using the patient’s arrival time at the emergency department as a point of reference. OR, odds ratio; CI, confidence interval; aOR, adjusted odds ratio; COVID-19, coronavirus disease 2019; SOFA, Sequential Organ Failure Assessment.

## Data Availability

All relevant data are within the manuscript and its Supporting Information files.

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
