# Peer review of "Impact of COVID-19 Pandemic on Management and Outcomes in Patients with Septic Shock in the Emergency Department"

_jpm, 2022, doi:10.3390/jpm12111803_

Round 1

Reviewer 1 Report

The study is well done and the paper is well presented. I have only few minor comments:

- the authors stated that several sepsis-related interventions were delayed in the post-COVID-19 group, as well as reported by other authors. Did you finf any relationship between the type of delayed intervention and sepsis outcome? For istance, delay in urological stone treatment could be associated with higher risk of sepsis and sepsis outcome.

- did you find any difference among the two groups in terms of antimicrobial bacterial resistance, if the data are available.

Author Response

C1) - the authors stated that several sepsis-related interventions were delayed in the post-COVID-19 group, as well as reported by other authors. Did you find any relationship between the type of delayed intervention and sepsis outcome? For istance, delay in urological stone treatment could be associated with higher risk of sepsis and sepsis outcome.

A1) We revised Table 4 and the Results section. Lactate measurements, blood culture, and source control were delayed in the post-COVID-19 group compared to the pre-COVID-19 group, although the administration of antibiotics and vasopressors was not statistically different between the groups. We conducted a multiple logistic regression analysis to determine factors associated with in-hospital mortality in patients with septic shock. Adjusting variables included three primary interventions: source control, antibiotic administration, and vasopressor administration. We repeated a multivariable analysis with a different reference to assess the relationship between delay in these three major interventions and the patient's prognosis. Antibiotic administration after three hours was significantly associated with an increase in in-hospital mortality, compared to antibiotic administration within three hours. However, neither delay in source control (≥12 h vs. <12 h) nor the administration of vasopressors (≥3 h vs. <3 h) was associated with in-hospital mortality. The changes made in the revised manuscript are highlighted in yellow.

C2) - did you find any difference among the two groups in terms of antimicrobial bacterial resistance, if the data are available.

A2) In the unmatched cohort, 8.9% (n = 329/3,697) of patients had resistant bacteria, and there was a significant difference between the pre-COVID-19 and post-COVID-19 groups (9.5% [n = 243/2,554] vs. 7.5% [n = 86/1,143]; p = 0.049). In the matched cohort, resistant bacteria were identified in 9.3% (n = 213/2,280), and there was a significant difference between the two groups (pre-COVID-19 group vs. post-COVID-19 group: 11.1% [n = 127/1,140] vs. 7.5% [n = 86/1,140]; p = 0.003). However, the presence of resistant bacteria did not have a significant association with in-hospital mortality in the univariable analysis, hence it was not included in the multivariable analysis (odds ratio 1.15 [95% CI 0.49-1.49], p = 0.281). The resistant bacteria data were added to Table 1. The changes made in the revised manuscript are highlighted in yellow.

Reviewer 2 Report

The authors tried to evaluate in-hospital mortality rate among ED sepsis patients before and after COVID pandemic by one multicenter database of prospectively collected data.

The study was restricted to the registered data. Providing information about sepsis/septic shock rate presenting to ED before or after COVID pandemic will be helpful. Besides, I also wonder that the rate or number of source control is similar before or after COVID pandemic. Would the authors add the valuable information to the result or limitation in the manuscript?

Author Response

The authors tried to evaluate in-hospital mortality rate among ED sepsis patients before and after COVID pandemic by one multicenter database of prospectively collected data.

C1) The study was restricted to the registered data. Providing information about sepsis/septic shock rate presenting to ED before or after COVID pandemic will be helpful. Besides, I also wonder that the rate or number of source control is similar before or after COVID pandemic. Would the authors add the valuable information to the result or limitation in the manuscript?

A1) The percentage of patients diagnosed with septic shock, defined as the Sepsis-3 criteria, is presented in Table 1. Source control among the study population was added to Table 1. In the matched cohort, the percentage of patients who received source control was significantly higher in the pre-COVID-19 group than in the post-COVID-19 group. The changes made in the revised manuscript are highlighted in yellow.

Reviewer 3 Report

This study aimed to determine if the pandemic had any influence on sepsis care in Korean Emergency Departments. The manuscript is well-written and its strength is data presentation illustrating the unmatched cohort and the propensity matched cohort. 

A few comments:

Section 2.1 "Despite some discrepancies in details," What are these discrepancies? Should provide more details or explanations. 

Section 2.2 I noticed the pre-COVID group spanned 2 years while the post-COVID group spanned for less than 1 year. Why not examine 2 years and how does this influence the results? How many Emergency Departments are involved in this registry? 

Section 3.1 How were the patients from each group paired matched? Tables and flow charts were nicely done. 

Section 4.0 The observation that sepsis-related interventions were delayed and bundle compliance was reduced but no effect on outcomes. Does this suggest that bundle is not specific and encompasses patients that are not that sick? 

"There is also societal pressure to visit the ED only when absolutely required" Not in my Emergency Department and probably not true in US. We have too many unnecessary visits. 

Author Response

C1) Section 2.1 "Despite some discrepancies in details," What are these discrepancies? Should provide more details or explanations. 

A1) Local institution situations demanded various modifications to the KCDC's recommendations for in-hospital infection control and prevention, despite the fact that participating institutions mostly adhered to these standards. As we could not consider it in this study, it was added in the limitation section.

“Finally, local institution conditions may have necessitated some adaptations to the KCDC's guidelines for in-hospital infection control and prevention, despite the fact that participating institutions mainly adhered to these guidelines. However, these factors were not considered in this study.”

We revised the Methods section additionally, as follows:

"There were differences in the available resources such as space and staffing at each local institution. Consequently, while participating institutions generally adhered to the KCDC's guidelines for in-hospital infection prevention, each institution had to respond optimally to the specific scenario it encountered."

Also, "Despite some discrepancies in details," was removed from the manuscript. The changes made in the revised manuscript are highlighted in yellow.

C2) Section 2.2 I noticed the pre-COVID group spanned 2 years while the post-COVID group spanned for less than 1 year. Why not examine 2 years and how does this influence the results? How many Emergency Departments are involved in this registry? 

A2) The Korean Shock Society Registry is an ongoing multicenter prospective study that includes 14 hospitals in South Korea. Due to the involvement of multiple hospitals, it takes considerable time for each hospital to input data, which is then rechecked and cleaned by the principal investigator to improve data reliability. We used the most up-to-date version of the data when designing our study. During the early phase of the COVID-19 pandemic included in our study period, our nation's quarantine regulations were at their most strict. It was during this time that the emergency department care process underwent the most significant changes. As sepsis patients are believed to have been most affected by the modified emergency department care process during this early phase of the COVID-19 pandemic, this may be the optimum period to compare sepsis patient management before and after the pandemic.

C3) Section 3.1 How were the patients from each group paired matched? Tables and flow charts were nicely done. 

A3) As stated in the Methods section, propensity score matching was used to adjust for imbalances between the two groups. Age, sex, comorbidities, severity measures (APACHE II score and initial SOFA score), initial lactate level, source of infection, and septic shock criteria meeting the Sepsis-3 consensus definition were used as the matching variables. We performed 1:1 matching using a caliper of 0.2. Statistical analysis was performed using SAS (version 9.4; SAS Institute, Cary, NC, USA) and R version 3.6.2 (Vienna, Austria; http://www.R-project.org/) by biostatisticians.

C4) Section 4.0 The observation that sepsis-related interventions were delayed and bundle compliance was reduced but no effect on outcomes. Does this suggest that bundle is not specific and encompasses patients that are not that sick? 

A4) It is unclear why there was no difference in in-hospital mortality between the two groups; however, this might be explained as follows. First, despite the delay in blood culture testing during the pandemic, there was no difference in the timing of antibiotic administration between the two groups. There was also no difference in the timing of the administration of vasopressors. Both are critical components in the treatment of septic shock. Even though lactate measurements were delayed, they did not appear to be clinically significant enough to impact the prognosis of the patient. Despite the fact that source control was delayed in the post-COVID-19 group, the multivariable analysis revealed that source control performed 12 hours after arrival at the emergency department was not associated with an increased in-hospital mortality rate compared to those who received source control within 12 hours. Second, even though the severity index of patients during the COVID-19 pandemic was higher than during the control period, it didn't seem to be severe enough to impact the outcomes. Finally, it appears that the number of ED patients declined during the pandemic, allowing medical professionals to devote more time and resources to treating critically ill patients. We revised the Discussion section. The changes made in the revised manuscript are highlighted in yellow.

C5) "There is also societal pressure to visit the ED only when absolutely required" Not in my Emergency Department and probably not true in US. We have too many unnecessary visits. 

A5) We agree with the reviewer. The sentence was removed from the manuscript.